# A PCR Test Using the Mini-PCR Platform and Simplified Product Detection Methods Is Highly Sensitive and Specific to Detect *Fasciola hepatica* DNA Mixed in Human Stool, Snail Tissue, and Water DNA Specimens

**DOI:** 10.3390/pathogens13060440

**Published:** 2024-05-23

**Authors:** Martha V. Fernandez-Baca, Alejandro Castellanos-Gonzalez, Rodrigo A. Ore, Jose L. Alccacontor-Munoz, Cristian Hoban, Carol A. Castro, Melinda B. Tanabe, Maria L. Morales, Pedro Ortiz, A. Clinton White, Miguel M. Cabada

**Affiliations:** 1Sede Cusco—Instituto de Medicina Tropical Alexander von Humboldt, Universidad Peruana Cayetano Heredia, Cusco 08002, Perurodrigo.ore@upch.pe (R.A.O.); maria.morales.f@upch.pe (M.L.M.); 2Division of Infectious Diseases, Department of Internal Medicine, University of Texas Medical Branch, Galveston, TX 77555, USA; alcastel@utmb.edu (A.C.-G.); mbtanabe@utmb.edu (M.B.T.);; 3Facultad de Ciencias Veterinarias, Universidad Nacional de Cajamarca, Cajamarca 06003, Peru; cahobanv@unc.edu.pe (C.H.); portiz@unc.edu.pe (P.O.); 4Universidad Peruana Cayetano Heredia-University of Texas Medical Branch Collaborative Research Center, Universidad Peruana Cayetano Heredia, Cusco 08002, Peru

**Keywords:** molecular diagnostics, *Fasciola hepatica*, mini-PCR, RT-PCR, ITS-1 gene

## Abstract

*Fasciola hepatica* has a complex lifecycle with multiple intermediate and definitive hosts and influenced by environmental factors. The disease causes significant morbidity in children and its prevalent worldwide. There is lack of data about distribution and burden of the disease in endemic regions, owing to poor efficacy of the different diagnostic methods used. A novel PCR-based test was developed by using a portable mini-PCR^®^ platform to detect *Fasciola* sp. DNA and interpret the results via a fluorescence viewer and smartphone image analyzer application. Human stool, snail tissue, and water samples were used to extract DNA. Primers targeting the ITS-1 of the 18S rDNA gene of *Fasciola* sp. were used. The limit of detection of the mini-PCR test was 1 fg/μL for DNA samples diluted in water, 10 fg/μL for *Fasciola*/snail DNA scramble, and 100 fg/μL for *Fasciola*/stool DNA scramble. The product detection by agarose gel, direct visualization, and image analyses showed the same sensitivity. The Fh mini-PCR had a sensitivity and specificity equivalent to real-time PCR using the same specimens. Testing was also done on infected human stool and snail tissue successfully. These experiments demonstrated that Fh mini-PCR is as sensitive and specific as real time PCR but without the use of expensive equipment and laboratory facilities. Further testing of multiple specimens with natural infection will provide evidence for feasibility of deployment to resource constrained laboratories.

## 1. Introduction

Fascioliasis is an emerging neglected zoonotic infection. The etiological agents are the trematodes *F. hepatica* and *F. gigantica*. *F. hepatica* is distributed worldwide and *F. gigantica* is limited to South/Southeast Asia and Africa. The burden of disease is unclear, but estimates ranging from 2.4–17 million individuals infected worldwide have been reported, associated to 35,206 disability-adjusted life years worldwide [1,2]. Over half of the human infections occur in the Andean altiplano of South America [3]. The Pan-American Health Organization (PAHO) has proposed the elimination of fascioliasis as a public health problem in the Americas by 2030 using a one health approach [4]. Efforts to decrease transmission of neglected tropical infections require accurate tools to determine the baseline epidemiology and to evaluate the impact of interventions on transmission. Methods to detect *Fasciola hepatica* in different intermediate and definitive hosts require significant training on a wide variety of parasitological methods [5]. However, most host specific methods available in endemic countries to detect fascioliasis have a suboptimal sensitivity even when performed by highly skilled technicians which is an important obstacle to elimination [5].

The Kato Katz test is recommended by the WHO for the diagnosis of high intensity *Fasciola* infections in humans [6]. This technique misses a third of the diagnosis and requires multiple samples to achieve an acceptable sensitivity [5,7]. Clinical scenarios in which egg counts are low, such as after treatment or annual mass drug administration, may decrease the sensitivity of Kato Katz even more [8]. Concentration methods such as Lumbreras sedimentation test and Mini-FLOTAC for human infection, and McMaster and Flukefinder^®^ for livestock infection have better performance than the Kato Katz but are more expensive, require training and devices, and may also be less sensitive in low burden infections [9,10]. *Fasciola* antigen detection in stool and antibody testing in bulk tank milk are commercially available for veterinary use in developed countries, but may be too expensive to use in endemic countries where subsistence family farming is practiced [11,12]. Serological methods using *Fasciola* E/S antigens are commercially available to diagnose acute and chronic infection in human. The specificity of serological tests in highly endemic areas is suboptimal and distinguishing between current or past infection may be difficult [5,6]. Lastly, microscopy and in vitro shedding assays are used to detect infection in snails, but these methods are not sensitive or specific.

Molecular diagnostic methods used to detect *Fasciola* DNA in different specimens have a high sensitivity and specificity. Quantitative real-time PCR methods can detect *Fasciola* DNA in human and animal stool, snail tissue, and water samples [13,14,15,16]. However, conventional RT-PCR require expensive equipment, highly trained personnel, and reliable power which may be lacking in endemic locations [5]. The ideal diagnostic method for resource constrained areas should be accurate, low cost, portable, and simple to perform and interpret. In this manuscript, we report on a PCR-based method using portable miniature thermocycler that can work on batteries to detect *Fasciola* DNA in stool, tissue, and water. This simplified PCR-based test for detection of *Fasciola* under field conditions may contribute to determine parasite burden and distribution.

## 2. Materials and Methods

### 2.1. Parasites and Biological Samples

*F. hepatica* adult parasites were obtained from naturally infected cattle livers that were discarded at a local slaughterhouse in Cusco, Peru. De-identified human stool samples positive or negative for F. hepatica eggs by microscopy and concordant serology by Fas2 ELISA were obtained from a biorepository maintained at −80 °C. DNA samples from *Hymenolepis nana*, *Trichuris trichiura*, *hookworm*, *Ascaris lumbricoides*, *Calicophoron microbothrioides*, *Taenia solium*, *Diphyllobothrium latum*, and *Echinococcus granulosus* were obtained from the same biorepository. Laboratory bred second generation *Galba truncatula* snails without *Fasciola* infection by microscopy were used for DNA extraction and testing (Figure 1).

### 2.2. DNA Purification

Extraction of DNA from adult *Fasciola* parasites was performed using the phenol-chloroform-isoamyl method. In brief, the flukes were washed with nuclease free water, cut into small pieces and resuspended in 500 µL of lysis buffer (Tris-HCl pH 7.5 10 mM, EDTA pH 8 10 mM, NaCl 50 mM and SDS 20% with 10 µL proteinase K and 20 µL β-mercaptoethanol). The tissue in the lysis buffer was incubated overnight at 56 °C. After incubation, an equal volume of phenol-chloroform-isoamyl alcohol (25:24:1) was added to the suspension to precipitate the proteins and the suspension was centrifuged at 14,000 RPM for 5 min. The resulting aqueous phase was carefully transferred to a new tube and an equal volume of propanol was added. The solution was gently mixed and then centrifuged at 14,000 RPM for 15 min. The DNA pellet at the bottom of the tube was washed three times with 70% ethanol and centrifuged at 14,000 RPM for 5 min. The final DNA pellet was air dried, resuspended in 100 µL nuclease free water, and stored at −20 °C.

A DNA purification from stool samples was performed using the E.Z.N.A^®^ Stool DNA kit (Omega Bio-Tek, Inc., Norcross, GA, USA) following the manufacturer’s instructions with the following modifications. Stool samples (200 mg) were placed in a 2 mL tube with 1.2 mL of lysis buffer, vortexed, and subjected to three cycles of freezing (−80 °C for 10 min) and heating (90 °C for 20 min). After this lysis step, the solution underwent DNA extraction according to the kit’s instructions.

A phenol-based method for purification of DNA from *Galba truncatula* with and without *Fasciola* infection was used [17]. The snails were extracted from their shells, homogenized in lysis buffer (2% (*w*/*v*) cethyl trimethyl ammonium bromide (CTAB), 1.4 M NaCl, 0.2% (*v*/*v*) β-mercaptoethanol, 20 mM EDTA, 100 mM Tris-HCl pH 8 and 0.1 mg/mL proteinase K), and incubated at 56 °C overnight. After incubation, the snail tissue underwent DNA purification using the phenol-chloroform-isoamyl alcohol (25:24:1) as described previously. The concentration and purity of the extracted DNA were determined using a NanoDrop 2000 UV–vis spectrophotometer (Thermo Scientific, Wilmington, NC, USA).

### 2.3. Real Time PCR Assays

Optimization for the reaction conditions for a real time PCR protocol was done, targeting the inter transcribed spacer-1 of the 18S rDNA gene of *F. hepatica* previously described [13]. The performance of this real time PCR test for *Fasciola* sp. served for comparison of the PCR using the mini-PCR platform. The limits of detection (LODs) and analytic specificity of the real time PCR reaction using the DNA of banked specimens were determined. To evaluate the LOD, DNA extracted from adult *Fasciola* as a template and the iTaq Universal SYBR-green Supermix (Bio-Rad, Hercules, CA, USA) for all PCR reactions was used. Each reaction contained 10 µL of the 2X SYBR Green Supermix, 1 μL of a 10 µM solution of each primer (F: 5′ ACC TGT ATG ATA CTC CGA TGG TAT GCT 3′, R: 5′ ACG TAT GGT CAA AGA CCA GGT TAT CAG 3′), 4 µL of template DNA, 1 µL of dimethyl sulfoxide, and distilled ultra-pure water for a final reaction volume of 20 μL. The real-time PCR was performed in a 96-well optical reaction plate using an Applied Biosystems 7500 Fast Real-Time PCR System (Applied Biosystems, Foster City, CA, USA). The PCR conditions included an initial incubation period at 50 °C for 3 min, denaturation at 98 °C for 5 min, and 40 cycles of 95 °C for 15 s, 64 °C for 30 s, and 72 °C for 30 s, followed by melting curve analysis. The initial incubation period was used to prevent dimer formation as primers they were transported at room temperature. Six non-template/DNAse free water reactions were tested as negative controls (NTC) in each run. Specificity of the products was confirmed with expected dissociation curves at 84 °C (Appendix A).

### 2.4. F. hepatica Mini-PCR Assay

*F. hepatica* DNA amplification was performed in a mini-PCR™ device (Minipcrbio, Cambridge, MA, USA) operated wirelessly using the mini-PCR application. The *F. hepatica* miniPCR reactions (up to 8 reactions per run) were assembled at room temperature in 0.2 mL tubes and then placed in the pre-programed thermocycler under the same optimized real time PCR protocol except for the melting curve analysis [13]. The end-point detection of the reaction products was performed at 35 to 40 amplification cycles until the optimal reaction conditions were selected. To test the limit of detection of the different *Fasciola* mini-PCR tests, decreasing concentrations of *Fasciola* DNA in water ranging from 1 ng/μL to 1 fg/μL were used. The same PCR primers targeting the inter transcribed spacer-1 of the *F. hepatica* 18S ribosomal gene used in the real time PCR were used [13]. The *F. hepatica* mini-PCR reaction was set to start with an incubation step at 50 °C for 3 min, denaturation at 98 °C for 5 min, and 35 to 40 cycles of 98 °C for 15 s, 64 °C for 35 s, and 72 °C for 35 s. The presence of a ~300 bp mini-PCR product was confirmed in all the reactions using DNA electrophoresis in a 1.6% agarose gels stained with ethidium bromide. The mini-PCR products were evaluated by visualization of the gels on an UV transilluminator.

### 2.5. Analysis of Mini-PCR Products Using the P51 Molecular Fluorescence Viewer

The results of the *Fasciola* mini-PCR reactions on positive and negative samples were observed via the P51™ molecular fluorescence viewer (Amplyus LLC, Cambridge, MA, USA). To determine the optimal cycle conditions for the *Fasciola* mini-PCR, the fluorescence of positive and negative samples tested was analyzed using the *Fasciola* mini-PCR and the results were compared via real time PCR based on SYBR-green ran on an Applied Biosystems™ 7500 Real-Time PCR System (Applied Biosystems, Foster City, CA, USA). The optimal cycle conditions were determined as the point in which we observed maximum fluorescence in the positive samples and minimal background fluorescence in the negative samples using the P51™ molecular fluorescence viewer. This point needed to match the point in which a lower CT value threshold was reached to detect the positive samples. The threshold of maximum fluorescence with minimal background was determined to be 35 cycles and all the subsequent mini-PCR experiments were ran using that number of cycles. After mini-PCR amplification, the unopened 0.2 mL tubes were immediately placed inside the P51™ molecular fluorescence viewer and the differences in color were evaluated by naked eye visualization. The interpretation was conducted distinguishing the green color fluorescent samples (positive samples) from the red-orange color controls (no DNA template/negative samples). The results were documented by the laboratory personnel and a picture of the tubes in the viewer was obtained for semi-quantitative image analysis.

### 2.6. Semi-Quantitative Colorimetric Analysis of Fasciola Mini-PCR Reaction Results

To quantify the fluorescence in the *Fasciola* mini-PCR products by colorimetric analysis, a free smartphone application called Prismo Mirage (Fuso Precision Co, Ltd., Kyoto, Japan) was used [18]. This application quantifies the intensity of light of a certain wavelength in images captured using the application or the phone camera. The differences in intensity are expressed in relative units of fluorescence. To determine the color intensity threshold to differentiate between positive and negative samples, *Fasciola* mini-PCR assays using different concentrations of *Fasciola* DNA were performed. The mini-PCR with no-template was also run, to determine the background color intensity. After the reaction, the mini-PCR tubes were placed in the P51 viewer in a dark room and a picture was obtained at a distance of 20 cm from the viewer. From the Prismo Mirage application, a picture for analysis was selected and the cursor was placed below the fluid level and in the center of the PCR tube to obtain the relative light units (RLU) recorded in the G (green) channel. All reactions were performed in triplicate and the histograms shown represent the mean ± SD of the relative fluorescent units of each condition repetition (Appendix A).

### 2.7. LOD Using Fasciola DNA Spiked in Water and Scrambled in Stool and Snail Tissue DNA

DNA extracted from adult *Fasciola* parasites was used to prepare eight serial dilutions in sterile water ranging between 1 ng/µL to 1 fg/µL. Four microliters of each dilution were used to run the mini-PCR and real time PCR reactions and compare their LOD. To evaluate if DNA from other sources interferes with the mini-PCR or real time reactions, *Fasciola* DNA at a concentration of 1 ng/μL was mixed with DNA extracted from *Fasciola* free human stool or snail tissue at a concentration of 10 ng/μL. The proportion of the mix was 1:10 for a final ratio of 1 ng of *Fasciola* DNA to 10 ng of stool or snail DNA. Serial dilutions of these specimens were tested using mini-PCR and real-time PCR to compare their performance. The products of the mini-PCR were detected using the P51 molecular fluorescence viewer by direct visualization and the colorimetric analysis using the Prismo Mirage application. Agarose gel electrophoresis was used to confirm the presence of a ~300 bp products in the mini-PCR reactions.

### 2.8. Specificity of the Real Time PCR and Mini-PCR

The analytical specificity of the mini-PCR and real-time PCR were evaluated by testing DNA samples extracted from *Hymnolepis nana*, *Trichuris trichiura*, hookworm, *Ascaris lumbricoides*, *Calicophoron microbothrioides*, *Taenia solium*, *Diphyllobothrium latum*, and *Echinococcus granulosus* with both tests. The specificity of the mini-PCR and real time PCR tests was analyzed by using DNA extracted from first-generation *Fasciola* free snails bred in the laboratory. Both tests were run in parallel by the same operator who was blinded to the identities and *Fasciola* infection status of the samples. Specificity reactions were repeated 10 times to show reproducibility with no amplification of PCR products with other parasites.

## 3. Results

### 3.1. Mini-PCR Products Detection without Post-Reaction Manipulation

The mini-PCR reaction was set up using *Fasciola* DNA at a concentration of 1 ng/μL for positive samples and DNAse free water for the negative controls. The reaction results were confirmed by agarose gel electrophoresis to demonstrate amplification products of the predicted size. A single amplicon of ~300 bp was detected by agarose gel electrophoresis only in samples containing DNA (*n* = 4), but not in the samples with no parasite DNA (non-DNA template) (Figure 2A). The mini-PCR products were analyzed by direct visualization with the P51 molecular fluorescence viewer. We confirmed the green color fluorescence in all *Fasciola* DNA positive samples but not in any of the no template or negative samples which showed a red-orange color (Figure 2B). The image analysis using the Primo Mirage application showed a difference in relative units of fluorescence between positive and negative samples (Figure 2C).

### 3.2. LOD of the Real Time PCR and Host DNA Interference

The analytical sensitivity of the real time PCR test was evaluated using serial dilutions of *Fasciola* DNA in water at concentrations ranging between 1 fg/μL–1 ng/μL (Appendix A). The LOD of the real time PCR test was 10 fg/μL of *Fasciola* DNA in water (Figure 3). To determine if the host DNA in stool samples or snail tissue interfere with the reaction, we evaluated the LOD of the real time PCR test using *Fasciola* DNA scrambled with DNA extracted from human stool or snail tissue at a ratio of 1:100. The real time PCR test using the *Fasciola*/human stool DNA scramble was able to detect 1 pg/μL of *Fasciola* DNA (Figure 3). Similarly, the real time PCR test using the *Fasciola*/snail DNA scramble was able to detect 1 pg/μL of *Fasciola* DNA (Figure 3). We did not detect amplification in samples with no *Fasciola* DNA (water with no template) and samples containing stool or snail tissue DNA only.

### 3.3. LOD of the Mini-PCR and Host DNA Interference

The analytical sensitivity of the mini-PCR test was evaluated using serial dilutions of *Fasciola* DNA in water at concentrations ranging between 1 fg/μL–1 ng/μL. The LOD of the Fh mini-PCR test was better than the real time PCR at 1 fg/μL of *Fasciola* DNA in water (Figure 4). Dilutions ≤ 0.1 fg were negative. All PCR product detection modalities showed a difference between the 1 fg/μL sample and the negative control, but the difference in agarose gel and P51 viewer was more clear than the Prismo Mirage image analyzer application (Figure 4). The mini-PCR test using the *Fasciola*/human stool DNA scramble was able to detect 100 fg/μL of *Fasciola* DNA. The mini-PCR products were detected with the same sensitivity by agarose gel electrophoresis (Figure 5A), P51 viewer, (Figure 5B), and the Prismo Mirage image analysis application (Figure 5C). Similarly, the mini-PCR test using the *Fasciola*/snail DNA scramble was able to detect 10 pg/μL of *Fasciola* DNA with all the product detection modalities (Figure 6). No amplification was detected in samples with no *Fasciola* DNA (water with no template) and samples containing stool or snail tissue DNA only. The specificity of the primers was confirmed by the expected dissociation curves at 84 °C (Appendix A).

### 3.4. Mini-PCR and Real Time PCR Specificity

The specificity of mini-PCR assay was evaluated using DNA from close related organisms. As expected, the agarose gel electrophoresis results showed ~300 bp amplicon only in the samples with F. hepatica DNA but not in DNA samples from *Hymenolepis nana, Trichuris trichiura*, hookworm, *Ascaris lumbricoides*, *Calicophoron microbothrioides*, *Taenia solium*, *Diphyllobothrium latum*, and *Echinococcus granulosus* (Figure 7). The visual detection of products by naked eye using the P51 molecular fluorescent viewer and the detection by image analysis using the smartphone application showed no cross-reaction with DNA of other parasites. The real time PCR did not show amplification of products with the DNA of the other parasites tested.

## 4. Discussion

Fascioliasis is a considered a neglected parasitosis and it is associated to significant morbidity particularly affecting children in the Andes altiplano [5]. The study of fascioliasis epidemiology requires the establishment of different methods to detect the parasite in a variety of specimens including human and animal stool, snails, and environmental samples. Many of these tests have suboptimal sensitivity, require highly skilled staff, and/or sophisticated equipment. The goal of this project was to evaluate the performance of a portable PCR machine for the diagnosis of fascioliasis. Our preliminary work demonstrated that real time PCR targeting the ITS-1 region of the *Fasciola* 18S rDNA gene could be a sensitive and specific tool to detect *Fasciola* DNA in human stool samples [13]. Our results confirmed the usefulness of real time PCR as a tool to detect *Fasciola* DNA in multiple specimens. In addition, the effectiveness of a simplified platform, the mini-PCR bio^®^ (Amplyus LLC, Cambridge, MA, USA) was demonstrated which could be used to potentially deploy molecular tools to areas without the resource to access and use expensive equipment. The mini-PCR machine is a portable thermocycler measuring 5 cm × 12 cm × 10 cm operated on AC power or batteries. The set up of the reactions is performed using a free software on a laptop or cell phone. Our mini-PCR showed equivalent results when compared to the real time PCR performed on an AB 7500 PCR system. The portability of the device and use of conventional PCR reagents and primers allow the use of this test in remote settings with limited infrastructure where fascioliasis is endemic.

Molecular methods are highly sensitive and specific tests, and require significantly less amount of sample for detection and avoid the need to differentiate between morphologically similar species. Additionally, these methods allow for early diagnosis and could potentially be applied to different samples and hosts [13,14,15]. PCR and real time PCR are considered the gold standard for molecular diagnostics [19]. PCR testing is not available in most resource constrain settings in part due to equipment costs and procurement capacity. The SARS-CoV2 pandemic uncovered these issues and pushed many countries to develop molecular testing capacity and infrastructure [20]. Although, access to advanced diagnostics remains limited in remote communities, the increase demand for reagents should allow for the establishment of a reliable supply chain [21,22]. In this scenario, conventional PCR reagents will likely be less expensive and more available. The mini-PCR uses conventional PCR primers and reagents. In contrast, loop mediated isothermal amplification (LAMP) requires the design of multiple pairs of primers and a specific DNA polymerase for isothermal amplification of DNA [23]. Similarly, the recombinase polymerase amplification (RPA) is another isothermal molecular test with a proprietary recombinase and polymerase combination limiting its widespread use [24].

The detection of amplification products in PCR based tests causes challenges in areas were training in molecular biology is limited. The lack of training may lead to contamination of PCR tests when manipulating reaction products. The mini-PCR does not require the post-PCR manipulation to determine if successful amplification has occurred. The mini-PCR yields a fluorescent product that is detected using a simple portable fluorescent viewer. The use of smartphone applications to determine the amount of fluorescence has been used successfully in other studies and also avoid the need to manipulate PCR products [25,26]. The use of image analysis for results interpretation decreases the subjectivity of direct visualization of the fluorescence specially in low burden infections. However, our mini-PCR showed consistent results with decreasing amounts of DNA even when naked eye interpretation was used. In other trematode infections, additional sample processing such as lyophilization increased test sensitivity [27]. We coupled the use of the P51 portable fluorescent viewer with a smart phone application designed to quantify color to interpret PCR results as a point-of-care (POC) test without opening the reaction tube. In our project, color quantification of pictures taken from P51 were useful to distinguish positive and negative samples with improved limits of detection and sensitivity than RT-PCR. In addition, the reading of the results allowed for closed tubed evaluation, as compared to PCR based lateral flow assays or conventional PCR electrophoresis. The use of product detection modalities that do not require post-PCR manipulation of samples decrease the possibility of cross contamination of the environments where tests are performed. This characteristic is very important in resource limited settings were training and decontamination reagents may be difficult to access.

*Fasciola* has a complex lifecycle, which requires a snail as the intermediate host, humans and primarily livestock as their final hosts [5,6,28]. In addition, environmental *Fasciola* DNA was isolated from water samples [16]. As the preferred diagnostic varies per host, a unified efficient testing method would ease the accurate assessment of the disease. In this project, the analytical sensitivity of mini-PCR^®^ in multiple samples (water, snail tissue, and stools of animals and humans) was higher than those obtained via RT-PCR under the same laboratory conditions (e.g., primers, PCR temperatures, amplification kits). This is consistent with other molecular diagnostics, such as RPA, which have shown increased sensitivity compared to the gold standard [13,19].

There are several limitations to our study. This study was a proof of concept and evaluated *F. hepatica* DNA mixed with stool/snail tissue DNA. The next step should be to evaluate the mini-PCR with clinical samples from different hosts (livestock, human, snails) to corroborate our findings and validate our results. Sample quality and DNA purification in field laboratories are barriers to deployment of molecular tests. We demonstrated the use of lysis-amplification methods (eg: IGEPAL) as an alternative to conduct DNA purification at the point of care in different pathogens [29]. The use of portable DNA mini-extractors showed feasibility to purify eDNA using simple filtration-concentration methods and could be used as an alternative [30]. Lastly, the use of smart phone applications for image analysis needs further validation of its sensitivity and specificity. In endemic countries, the accessibility to smartphones could limit the widespread implementation of the diagnostics. Furthermore, despite being minimal, some training may be required for the use of the phone application.

In this manuscript, we have demonstrated that a PCR test using the mini-PCR platform performs as well or better than one real time PCR using a conventional thermocycler. This mini-PCR was sensitive and specific for the detection of *Fasciola* DNA in different conditions. The WHO recommends that point of care diagnostics should meet the ASSURED criteria (affordable, sensitive, specific, user-friendly, rapid, equipment-free, delivered) [31]. Creating an all-host accurate diagnostic test for *Fasciola* will likely need the use of portable molecular diagnostics our future work will focus on evaluating the mini-PCR in clinical and environmental samples.

## Figures and Tables

**Figure 1 pathogens-13-00440-f001:**
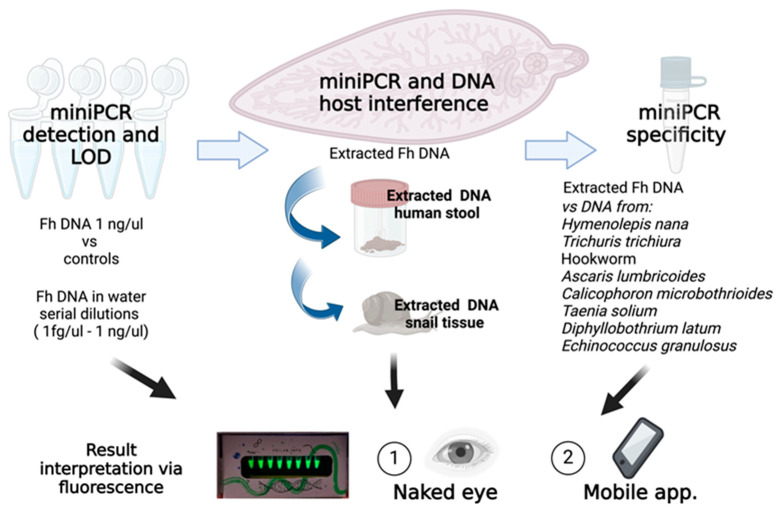
“Schematics of miniPCR^®^ standardization as a proof of concept” (Illustration created with BioRender.com, accessed on 21 May 2024).

**Figure 2 pathogens-13-00440-f002:**
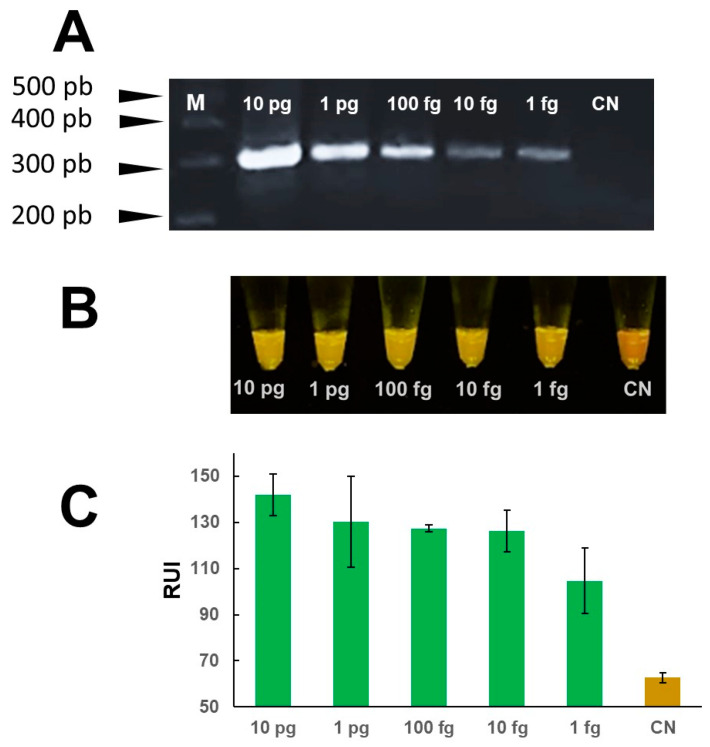
“Amplification of *Fasciola* DNA with the mini-PCR”. Purified DNA [1 ng/µL] from *F. hepatica* was used as template to amplify a ~300 bp sequence in the ITS-1 region of the *Fasciola* 18S rDNA gene using the mini-PCR. (**A**) A ~300 bp amplicon was detected in agarose gels in all the positive samples. (**B**) The color difference between positive and negative samples was visualized in the P51 molecular fluorescence viewer. (**C**) The colorimetric analysis using the Prismo Mirage application determined a difference in relative fluorescence units in positive (*n* = 4) and non-template or negative control (*n* = 4) samples.

**Figure 3 pathogens-13-00440-f003:**
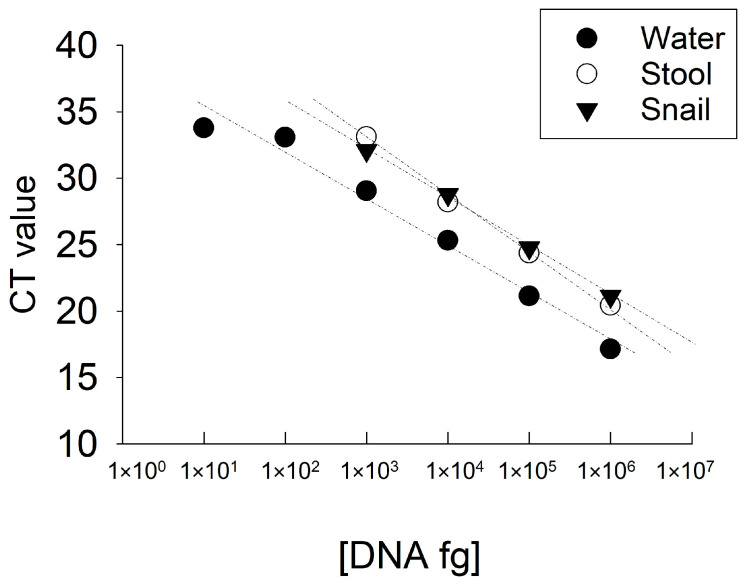
“Limit of detection of SYBR Green based real time PCR assay targeting the ITS-1 region of the *Fasciola* 18S rDNA gene using serial dilutions of *Fasciola* DNA in water or scrambled with DNA extracted from human stool or snail as template.” The LOD of *Fasciola* DNA against the reaction cycle threshold (CT) values in water (triangle), stool (empty circle) and snail tissue (black circle) specimens are shown in the graph.

**Figure 4 pathogens-13-00440-f004:**
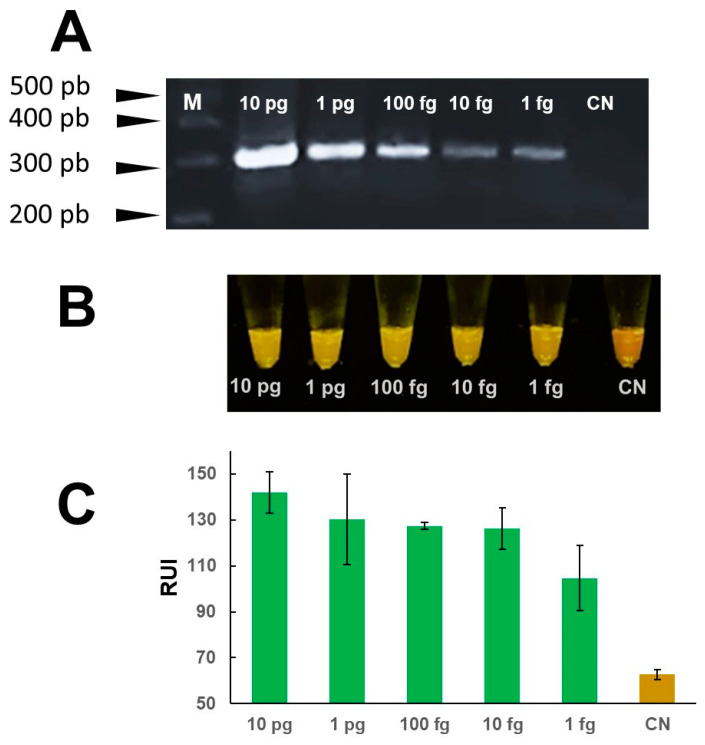
“LOD mini-PCR assay to detect *Fasciola* DNA in water”. Serial dilutions of *Fasciola* DNA at concentrations between 10 pg/μL–1 fg/μL amplified ~300 bp products in mini-PCR. Amplicons were detected by agarose gel electrophoresis (**A**) and by direct visualization using the P51 molecular fluorescence viewer (**B**) with a LOD of 1 fg/μL. Colorimetric analysis with the Prismo Mirage application (**C**) showed a difference in relative fluorescence units between the no template control and the 1 fg/μL concentration.

**Figure 5 pathogens-13-00440-f005:**
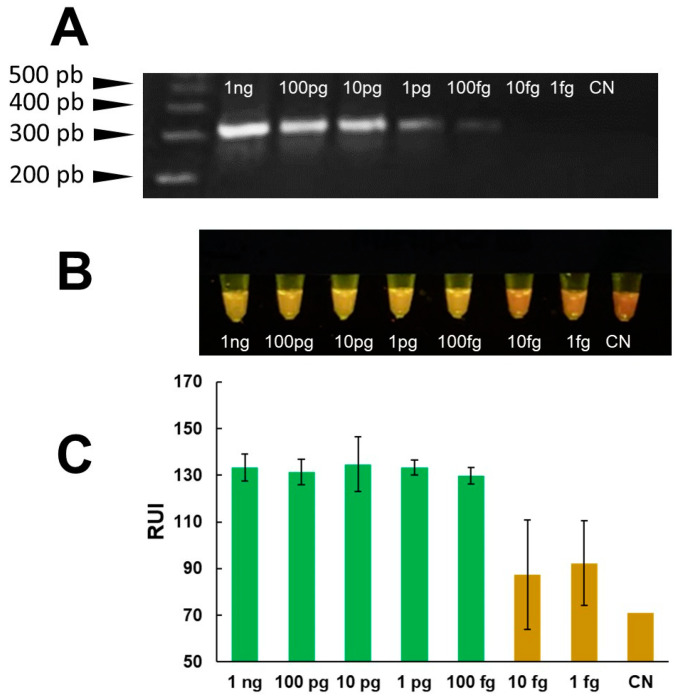
“LOD of mini-PCR assay for the detection of *Fasciola* DNA scrambled with stool DNA”. Serial dilutions of the scrambled DNA equivalent to a *Fasciola* DNA concentrations between 1 ng/μL–1 fg/μL were used to determine if stool DNA interfered with the reaction. A ~300 bp amplicon was detected in agarose gel electrophoresis (**A**) and by direct visualization using the P51 molecular fluorescence viewer with a LOD of 100 fg/μL (**B**). The colorimetric analysis with Prismo Mirage application (**C**) showed a difference in relative fluorescence units between the 100 fg/μL sample and the no template control.

**Figure 6 pathogens-13-00440-f006:**
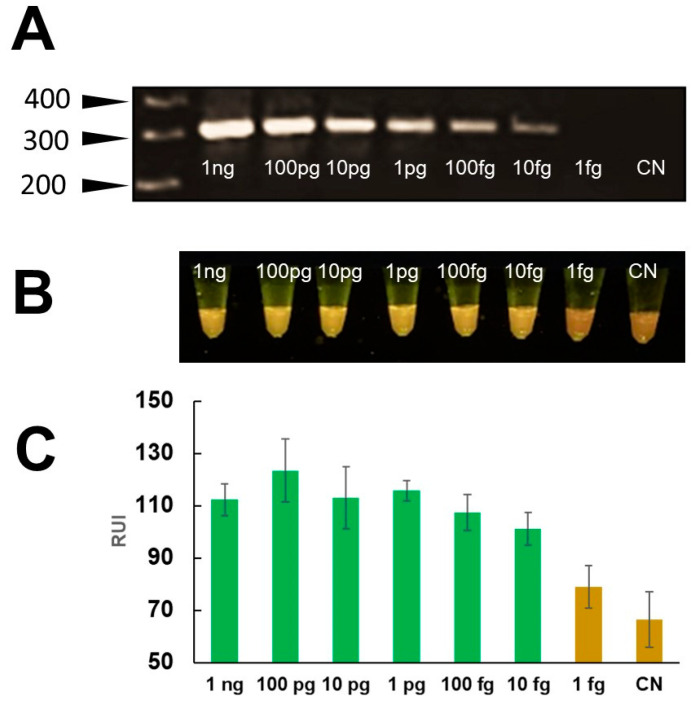
“LOD of mini-PCR assay for the detection of *Fasciola* DNA scrambled with snail DNA”. Serial dilutions of the scrambled DNA equivalent to *Fasciola* DNA concentrations between 1 ng/μL−1 fg/μL were used to determine if snail DNA interfered with the reaction. A ~300 bp amplicon was detected by agarose gel electrophoresis (**A**) and by direct visualization using the P51 molecular fluorescence viewer with a LOD of 10 fg/μL (**B**). The colorimetric analysis with Prismo Mirage application (**C**) showed a difference in relative fluorescence units between the sample with 10 fg/μL and the no template control.

**Figure 7 pathogens-13-00440-f007:**
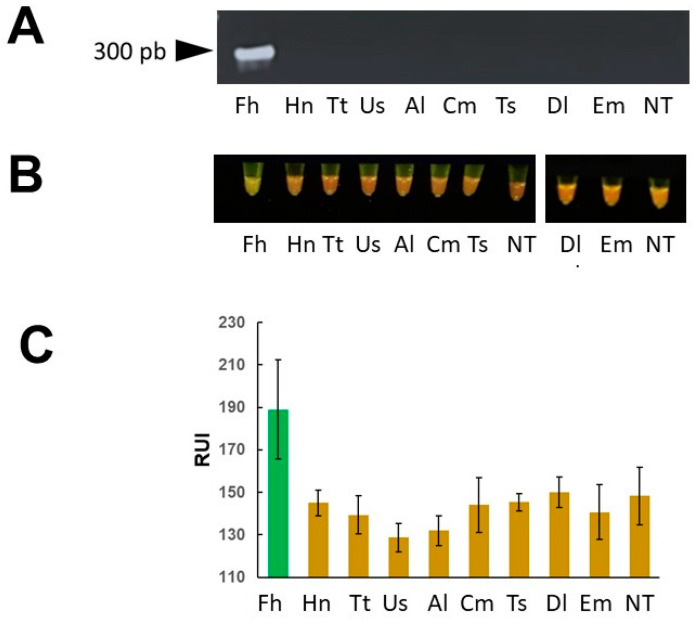
“Specificity of mini-PCR assay”. *Fasciola* and other helminths DNA at a concentration of 1 ng/μL was used as template in the mini-PCR reactions. A ~300 bp product was detected in agarose gel electrophoresis (**A**) only in the *Fasciola* DNA containing sample and in none of the other helminth DNA samples. The naked eye visualization with the P51 molecular fluorescence viewer (**B**) and image analysis using the Prismo Mirage application (**C**) showed positive results only in the sample containing *Fasciola* DNA. *Fasciola hepatica* (Fh), *Hymenolepis nana* (Hn), *Trichuris trichiura* (Tt), hookworm (Ac), *Ascaris lumbricoides* (Al), *Calicophoron microbothrioides* (Cm), *Taenia solium* (Ts), *Diphyllobothrium latum* (Dl), and *Echinococcus granulosus* (Em), no template (NT).

## Data Availability

The raw data used for this manuscript is available at the Harvard Dataverse public repository (https://dataverse.harvard.edu), accessed on 28 April 2024. The database without identifiable information can be requested via this repository.

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
