# Peer review of "A PCR Test Using the Mini-PCR Platform and Simplified Product Detection Methods Is Highly Sensitive and Specific to Detect Fasciola hepatica DNA Mixed in Human Stool, Snail Tissue, and Water DNA Specimens"

_pathogens, 2024, doi:10.3390/pathogens13060440_

Round 1
Reviewer 1 Report
Comments and Suggestions for Authors
The study established a mini-PCR platform for the detection of Fasciola hepatica DNA in multiply samples, which enriches the diagnostic methods of Fasciola hepatica. However, there are some aspects need to be addressed.
1. Title: to detect DNA from which samples? This should be indicated in the title.
2. Abstract: as indicated by the authors at the end, at least a small scale of samples should be applied to evaluate the method.
3. Introduction and other parts: Refs in the MS should be placed before the dot, but not after. e.g., Line 41-.[1-2].
4. Methods:
4.1 As to "F. hepatica Mini-PCR assay", did the authors sequenced the band with ~ 300 bp? This shoud included to confirm the band was true.
4.2 Did the author repeat tests for specificity and sensitivity?
5. Results:
5.1 As to "3.2. LOD of the real time PCR and host DNA interference",melting curves should be included, only CT values are not sufficient.
5.2 As to "3.3. LOD of the Fh mini-PCR and host DNA interference", the DNA needs to be further diluted to 0.1 fg, 0.01 fg and so on.
5.3 A small scale of clinical samples from water, stool and snail need to be tested in both methods to compare those methods.
6. Discussion:
As to the second paragraph, the authors compared different diagnostic methods. However, the authors only mentioned the weakness of other methods. Advantages of other methods are also needed to comprehensively compare those methods with the established method in this study.
7. References:
The format was inconsistent. Re-checked them.
Comments on the Quality of English LanguageMinor editing of English language required
Reviewer 2 Report
Comments and Suggestions for Authors
Dear Editor,
The manuscript by Fernandez-Baca and others, A PCR test using the mini-PCR platform and simplified product detection methods is highly sensitive and specific to detect Fasciola hepatica DNA describes the use of a portable mini-PCR® platform to detect Fasciola sp. DNA and results interpretation via a fluorescence viewer and smartphone image analyzer application in the development of portable tools for fascioliasis diagnosis. The authors made use of human stool samples, snail tissue, and water samples to extract DNA, and also evaluated diagnostic parameters for the tests including LOD, sensitivity, and specificity.
I have carefully read through the manuscript and have these comments to make to improve on understanding and add clarity.
1. It would be important for the authors to provide information on the burden exerted by the disease (mortality, morbidity, etc) both in the abstract and in the introduction. In addition, it will be important to mention Fasciola gigantica in the introduction even if this manuscript focuses on Fasciola hepatica
2. The authors should make use of the passive voice in the abstract and methodology.
3. I am concerned about whether any ethical clearances were needed for this work and if these were obtained.
4. The authors should use the proper style of writing scientific names and also referencing for this journal.
5. In line 147, the authors should provide a rationale for the extra incubation step at the beginning of the PCR.
6. The authors should provide the sizes for molecular weight markers used in the agarose gel electrophoresis
7. It would be a good idea to begin the discussion by re-introducing the problem and restating goal of this project.
8. While the project talks about the application of the developed test in low-resource settings, the authors should also indicate that some training will still be required for the application of this method and smartphones may not always be available.
Comments on the Quality of English LanguageMinor editing of English language required
